# Extracellular miRNAs as Biomarkers of Head and Neck Cancer Progression and Metastasis

**DOI:** 10.3390/ijms20194799

**Published:** 2019-09-27

**Authors:** Zuzanna Nowicka, Konrad Stawiski, Bartłomiej Tomasik, Wojciech Fendler

**Affiliations:** 1Department of Biostatistics and Translational Medicine, Medical University of Lodz, 92-215 Lodz, Poland; zuzanna.nowicka@stud.umed.lodz.pl (Z.N.); konrad@konsta.com.pl (K.S.); bartlomiej.tomasik@umed.lodz.pl (B.T.); 2Postgraduate School of Molecular Medicine, Medical University of Warsaw, 02-091 Warsaw, Poland; 3Department of Radiation Oncology, Dana-Farber Cancer Institute, Boston, MA 02115, USA

**Keywords:** miRNAs, biomarker, HPV, head and neck cancer, radioresistance, extracellular, serum, plasma, saliva

## Abstract

Head and neck squamous cell carcinomas (HNSCCs) contribute to over 300,000 deaths every year worldwide. Although the survival rates have improved in some groups of patients, mostly due to new treatment options and the increasing percentage of human papillomavirus (HPV)-related cancers, local recurrences and second primary tumors remain a great challenge for the clinicians. Presently, there is no biomarker for patient surveillance that could help identify patients with HNSCC that are more likely to experience a relapse or early progression, potentially requiring closer follow-up or salvage treatment. MicoRNAs (miRNAs) are non-coding RNA molecules that posttranscriptionally modulate gene expression. They are highly stable and their level can be measured in biofluids including serum, plasma, and saliva, enabling quick results and allowing for repeated analysis during and after the completion of therapy. This has cemented the role of miRNAs as biomarkers with a huge potential in oncology. Since altered miRNA expression was described in HNSCC and many miRNAs play a role in radio- and chemotherapy resistance, cancer progression, and metastasis, they can be utilized as biomarkers of these phenomena. This review outlines recent discoveries in the field of extracellular miRNA-based biomarkers of HNSCC progression and metastasis, with a special focus on HPV-related cancers and radioresistance.

## 1. Introduction

miRNAs are short non-coding RNA molecules that post-transcriptionally regulate gene expression by binding to the 3′-untranslated regions (3′UTR) of the target mRNAs [1,2]. The complementarity of this binding, determined by the 8 nucleotides long ‘seed region’ at the 5′ end of mature miRNA, dictates the fate of targeted mRNAs, as follows: Either they are degraded or their translation is inhibited [3]. Over 2600 mature miRNA sequences are registered in the current miRBase release (miRBase 22) and over 60% of human protein-coding genes are estimated to be regulated by miRNAs [4]. Owing to the redundancy of miRNAs and their targets (not only is one gene regulated by multiple miRNAs that bind to its’ 3′UTR, but also one miRNA may target multiple mRNAs), the regulatory network of miRNAs and protein-coding genes is complex and influenced by changes in the cell’s transcriptional program.

miRNAs influence multiple biologic processes, including ones that constitute the hallmarks of cancer, such as proliferative signaling, cell death, and metastasis [5,6,7]. Their expression profile also varies under changing biological conditions, including pathologic states, making it possible to infer information about these states based on miRNA abundance. Although they exert their regulatory function inside cells, extracellular miRNAs were also detected in various body fluids, including blood serum [8] and plasma [9], urine [10], saliva [11], semen [12], and milk [13]. Altered miRNA expression profile as a result of the cancerous process can, therefore, be detected not only in the tumor biopsy, but also in easily accessible biospecimens like peripheral blood and—especially relevant in HNSCC—saliva, making them minimally invasive biomarkers.

On top of the accessibility, the extreme stability of miRNAs strengthens their biomarker potential, spawning nearly 20,000 publications over the last two decades. miRNAs, including extracellular ones, are highly resistant to degradation by extracellular nucleases [14]. Studies performed in fresh frozen, formalin-fixed paraffin-embedded (FFPE), and serum samples stored for many years provided evidence that miRNA expression is stable and largely unaffected by storage conditions [15,16], even in degraded RNA preparations [17]. At first, it was assumed that the stability of miRNAs in biofluids stems from their protection by membrane-bound vesicles, such as exosomes. Surprisingly, it turned out that most circulating miRNAs are not encapsulated in vesicles, but associated with protein complexes containing Argonaute 2 (Ago2), a key effector of miRNA-mediated silencing [18].

The identification of extracellular Ago2-miRNA complexes prompted researchers to suggest that a functional miRNA-induced silencing complex released into circulation may regulate gene expression in recipient cells, providing a means of cell–cell communication [18]. This hypothesis was reinforced by reports that miRNAs protected by vesicles can be transferred between cells, influencing their gene expression and provoking biologic effects [19,20]. Opposed to this theory, it was argued that miRNAs are merely cellular byproducts released passively during apoptosis and necrosis [14,21], due to their extremely low levels in circulation and biologically irrelevant number of miRNAs in most exosomes derived from standard preparations [22].

Regardless of their actual function in cellular signaling, the potential of extracellular miRNAs in cancer diagnosis, prognosis, and therapy are well established [23,24,25]. Compelling evidence suggests that their deregulated expression profile reflects pathologic states associated with tumor initiation, progression, and metastasis. Consequently, their use as biomarkers in the clinical setting may help predict patient survival, locoregional relapse, and distant metastases, or to inform clinical decisions based on the predicted responsiveness to a specific form of treatment. Such biomarkers would be particularly useful in head and neck squamous cell carcinomas (HNSCCs), a highly heterogeneous group of tumors that are one of the predominant causes of cancer-related deaths worldwide [26]. They occur in 800,000 patients every year and the incidence in many countries is rising, mostly due to the increasing number of human papillomavirus (HPV)-mediated cancers [27,28].

Despite multiple treatment options encompassing surgery, radiotherapy, chemotherapy, and, more recently, immunotherapy [29,30], approximately 50% of treated patients experience disease recurrence and do not respond to subsequent therapeutic interventions [31]. Locoregional recurrences are the most frequent, followed by distant metastases. In addition, second primary cancers in the head and neck region and other sites (e.g., lungs, upper gastrointestinal tract) occur at an annual rate of approximately 2–3%, particularly in patients with HPV-negative non-oropharyngeal cancers [31].

Although all HNSCC tumors derive from epithelial cells, they are molecularly heterogeneous and the histological or clinical stages do not correlate with clinical response and prognosis in advanced HNSCC [32], posing the need for other prognostic and predictive biomarkers. One such marker already established in everyday routine is HPV status. It is, however, only useful in cases of oropharyngeal cancer, where it is associated with a better prognosis due to the increased radiosensitivity of HPV-associated tumors [33]. Recent efforts have therefore been focused on finding more universal and easily accessible prognostic and predictive biomarkers that would drive therapeutic choices in HNSCC and lead to overall survival improvement, such as extracellular miRNAs.

## 2. Extracellular miRNAs as Biomarkers of HNSCC Metastasis and Progression: Progress and Recent Discoveries

Up to date, several miRNAs expressed in plasma, serum, or saliva were reported as potentially useful in the prognosis of head and neck cancer or in predicting metastases and response to therapy (Figure 1, Table 1).

miR-21 is one of the few miRNAs for which expression has been consistently reported to be higher in the plasma, serum-derived exosomes, and in the whole blood of patients with head and neck cancers, compared to healthy controls [46,52]. The higher expression has been also linked to a more advanced stage, less differentiated oral cancer histology and positive lymph node status in laryngeal squamous cell carcinoma and oral squamous cell carcinoma [46,50]. This is consistent with the results from many studies linking high miR-21 expression to chemoresistance, metastasis, and worse survival in cancer patients [5]. Another interesting report concerning miR-21 comes from a study by Hsu et al., who measured plasma levels of ten miRNAs in 50 HNSCC patients and reported that the levels of miR-21 and miR-26b were reduced post-surgery in the plasma of HNSCC patients with good prognosis (patients who survived for more than 1 year after surgery) and remained high in patients who died within one year after surgery [52]. This result was concordant with the fact that miR-21 expression level was upregulated both in the plasma from HNSCC patients and in the HNSCC tissues. Given, however, that only 3 patients were included in the ‘poor prognosis’ group, it is not unexpected that no significant changes in these miRNA expression levels were detected in these patients.

miR-21 is a known “oncomir” (miRNA with oncogenic properties) with high expression in the solid tissues of different tumors [56]. Ultimately, its increased levels were also reported in multiple non-oncologic states, including sepsis-associated cardiac dysfunction [57], impaired graft function in renal transplant recipients [58], and traumatic brain injury [59]. This is not surprising, given the inherent redundancy of miRNAs and their target genes, but casts doubt upon the use of a single miRNA as a sufficiently specific biomarker for outcome-predictive purposes. On the other hand, plasma levels of miR-21 were also reported to decrease in plasma after tumor excision in patients with HNSCC by two independent groups of researchers [52,53], pointing to its’ potential utility as a marker for disease monitoring.

miR-9 is another miRNA with consistent reports from different studies. Its lower expression in plasma was observed at recurrence or metastasis in nasopharyngeal carcinoma and in metastatic patients compared to patients without distant metastases. In a study by Sun et al., including 104 patients with oral squamous cell carcinoma (OSCC), researchers reported downregulated miR-9 expression in sera of patients with OSCC compared to healthy controls and the association between decreased miR-9 expression and poor prognosis, more advanced stage and lymph node metastases [44]. These results are concordant with decreased miR-9 expression reported in OSCC tissues and oral cancer cell lines by other authors and may indicate its role as a tumor suppressor and regulator of OSCC progression.

Seemingly, an even lower consistency between studies can be observed in case of miR-31, for which levels were reported by Liu et al. to be higher in the plasma of patients with OSCC than in healthy controls and was reduced following tumor excision [42]. The same researchers reported previously that miR-31 was significantly upregulated in OSCC tissues and that it mediated oral oncogenesis by hypoxia pathway regulation [60]. Meanwhile, Yi et al. more recently reported that miR-31 level was lower in the whole blood of patients with nasopharyngeal carcinoma (NPC) than in healthy controls and that its reduced expression was associated with local lymph node metastases [37]. In the same study, lower miR-31 expression was associated also with more advanced disease stage and TNM classification. The results of miRNA profiling in different tumor sites and, more importantly, in different biological fluids, however, should be interpreted with caution and probably should not be directly compared.

Plasma is the liquid fraction of the blood yielded from anti-coagulated blood. Coagulated blood yields serum, of which the composition is similar to that of plasma, excluding fibrinogen and other proteins used in blood clotting. Blood cells are an important source of circulating miRNAs [61] and they might release RNA, including miRNAs, during the coagulation process [62]. One should therefore not necessarily expect concordant miRNA expression profiles between the whole blood, plasma, serum, and serum-derived exosomes, especially given the selective miRNA packing to exosomes, which has also been described in HNSCC [63].

miRNAs originating from the cancerous tissue are not only attractive candidates for diagnostic biomarkers, but also for clinical use to monitor disease progression and predict recurrence. Tumor origin is certainly more plausible in case of miRNAs, of which expression is higher both in the solid tumor and biofluid samples. As an example, Wong et al. reported that miR-184 levels were higher in tongue squamous cell carcinoma (TSCC) samples than in normal tissues and in the plasma of patients with tongue SCC [48]. More convincingly, miR-184 levels were also associated with the presence of a primary tumor and decreased significantly following surgical resection, highlighting not only its diagnostic utility but also its potential in monitoring the disease and recurrence. In another recent study, Shi at all analyzed 260 serum samples from patients with OSCC with the goal to elucidate the role of miRNAs as prognostic biomarkers [45]. The resulting two-miRNA signature, based on the serum expression of miR-626 and miR-5100, was an independent predictor of patient survival. The researchers also measured serum expression of these miRNAs in paired pre- and post-operative specimens from another 40 patients with OSCC. Not only were serum levels of both miRNAs decreased significantly in post-operation samples, but the reduction was also positively correlated with tumor volume, pointing to the tumor as the source of miR-626 and miR-5100.

Saliva is a biofluid composed of multiple enzymes, hormones, antibodies, antimicrobial proteins, and cytokines. It is also an inexpensive and easily accessible source of biomarkers for HNSCC prognosis and disease monitoring. As has already been mentioned, miRNAs are detectable in saliva [11], where they may be released from the blood or from local normal or cancerous cells, either via cell death or excretion in exosomes and microvesicles [64]. Given its high potential utility in the clinical setting, as the sampling does not require drawing peripheral blood, in an increasing number of studies researchers are focused on identifying salivary miRNAs exclusively associated with the presence of cancer. Liu et al. reported a significant elevation of miR-31 in both saliva and plasma of OSCC patients (n = 43) compared to age and sex-matched controls [42]. The researchers noted not only that miR-31 levels reduced significantly following surgical removal of the primary tumor, potentially pointing to the tumor tissue as its primary source, but also that the miR-31 level was higher in saliva than in plasma, both in the patients and controls. This result is unsurprising given the direct proximity of OSCC to the sampling site and suggests that, in some patients, saliva might be a preferential source of biomarkers.

Other studies, with a similar design aimed to identify salivary miRNAs associated with the presence of cancer, were also performed in other groups of patients. Duz et al. profiled miRNA expression in saliva samples from 25 patients with TSCC, a subset of OSCC, and 25 controls. The researchers identified miR-139 as a potential biomarker based on its remarkably decreased expression in samples from tongue SCC patients compared to controls [51]. Importantly, miR-139 expression reverted to normal following surgery and its level could be used to discriminate pre- and post-operative TSCC patients. These results are consistent with miR-139’s established role as a tumor suppressor [65,66].

Greither et al. aimed to detect miRNAs in salivary samples collected post-radiation therapy from HNSCC patients [55]. They measured the levels of ten HNSCC- or radiation-associated miRNAs selected from the literature pre, during, and post-radiotherapy (RT) and concluded that miR-93 and miR-200a have a significantly higher expression 12 months after the completion of RT compared to the baseline. Although no firm conclusions can be driven from such study, as neither the tumor source of these miRNAs or association with RT were investigated, it might point to the role of miR-200a and miR-93 in tumor suppression or the healing process following RT and their biomarker potential.

Therefore, because of substantial heterogeneity of miRNA expression reported by different studies, it may be reasonable to assume that finding ones associated directly with prognosis may be difficult, if not impossible. However, alternative options of using microRNA biomarkers in guiding therapy would be to identify the ones that are predictive for immediate treatment effects (i.e., radiotherapy) or associated with a major emerging factor determining prognosis (e.g., HPV infection status).

## 3. Extracellular miRNAs Associated with HPV Infection and Other Risk Factors in HNSCC

HPV-associated HNSCCs have distinct molecular and clinical features that differentiate them from HPV-negative cancers [67]. They are more often localized in the oropharynx, less histopathologically differentiated, and associated with better prognosis, most likely due to the higher radio- and chemosensitivity of HPV-infected epithelial cells. Some features of HPV-associated cancers were suggested to result from the viral modulation of host miRNA expression [68].

It was shown that HPV status may impact miRNA expression patterns in HNSCC [69]. Salazar-Ruales et al. found that salivary miRNAs, including miR-122, miR-124, miR-205, and miR-146a, are downregulated in HPV-associated HNSCC and have the potential to differentiate between HPV+ and HPV− cancers [70]. In a similar study, Wan et al. compared salivary miRNA expression between HPV-positive and HPV-negative HNSCC patients and found that a panel consisting of miR-9, miR-134, miR-196b, miR-210 and miR-455 could discriminate HPV-positive HNSCC from HPV-negative HNSCC [71]. In this study, however, only a panel of selected 9 miRNAs was evaluated, preventing a comprehensive comparison with results of other studies.

Peacock et al. evaluated the miRNA expression profile of small extracellular vesicles (EVs) released by HPV(+) and HPV(−) OPSCC cell lines [72]. They found that 14 miRNAs were upregulated and 19 were downregulated in EVs derived from HPV+ cell lines. Among the 14 upregulated miRNAs were 3 miRNAs from the miR-9 family (miR-9-1, miR-9-2, and miR-9-3), suggesting that the upregulated miR-9 levels in saliva of HNSCC patients reported by Wan et al. could potentially derive from EVs released from the tumor.

Smoking and alcohol consumption are well-established risk factors for HNSCC. Several authors of the studies listed in Table 1 investigated the relationship between these factors and the levels of extracellular miRNAs associated with prognosis and metastasis in HNSCC. No study, however, found any associations of miRNA levels with smoking status [44,46] or excessive alcohol intake [45].

## 4. Radiotherapy-Associated Extracellular miRNAs in HNSCC

Radiation therapy is one of the main treatment modalities in HNSCC. The most important cause of disease-related mortality is a locoregional recurrence, which is strongly associated with tumor-specific radioresistance. Since the biological effect of RT differs between patients, depending on the primary site of cancer, HPV involvement in tumorigenesis, and the individual radiosensitivity of each cancer, substantial efforts are being made to discover the mechanisms that govern tumor sensitivity to radiation and to identify biomarkers predictive of the response to RT. miRNAs are involved in biological processes relevant to the cellular response to radiation, including DNA damage response [73], apoptosis [74], and tumor adaptation to hypoxic environments [75]. In this regard, circulating miRNAs have been proposed as stable and accurate markers of tumor response to irradiation and normal-tissue toxicity [76].

Summerer et al. analyzed miRNA expression profiles in the plasma samples of 18 HNSCC patients undergoing radiochemotherapy. They found that six miRNAs (miR-425-5p, miR-21-5p, miR-106b-5p, miR-590-5p, miR-574-3p, miR-885-3p) were downregulated after two days of treatment, pointing to their potential use as radiation-responsive biomarkers [77]. Noteworthy, the researchers tested the hypothesis that these upregulated miRNAs originated from peripheral blood mononuclear cells (PBMCs) and they found no correlations between miRNA expression in PBMCs and plasma. However, the same miRNAs were upregulated after in vitro radiochemotherapy in primary HNSCC cell cultures, from which the researchers concluded that they might be related to therapy effects on tumor cells. It remains unclear why the plasma expression of miRNAs released from the tumor would be lower after radiochemotherapy.

In a follow-up study, the same authors evaluated blood plasma miRNAs in a validation cohort of 11 HNSCC patients treated with radiotherapy [54]. They found that high plasma levels of miR-186-5p, miR-374b-5p, and miR-574-3p prior to treatment were associated with shorter progression-free or overall survival and that high plasma levels of miR-28-3p, miR-142-3p, miR-191-5p, miR-195-5p, miR-425-5p, and miR-574-3p after treatment were associated with a worse prognosis. None of the miRNAs, however, showed consistent change after treatment. Although all the therapy-responsive miRNAs in plasma were also detectable in the patients’ FFPE tumor samples, the expression levels in plasma and tissues were not correlated.

Other miRNAs suggested as prognostic and predictive biomarkers for HNSCC were also associated with tumor response to RT. miR-296, for which higher expression has been described in plasma of oropharyngeal cancer patients with lymph node metastases [38], has been also suggested to regulate the radiosensitivity in laryngeal squamous cell carcinoma (LSCC) by targeting *MDR1* gene [78]. Its higher expression in the tumor tissue was also associated with radioresistance and recurrence in early-stage LSCC. miR-93 expression in saliva was reported to be upregulated after radiotherapy in HNSCC patients [55]. Since its expression was found to be downregulated in radioresistant NPC cells [79], it may also have potential as a biomarker of radiosensitivity.

## 5. miRNAs as Predictive Biomarkers for Immunotherapy in HNSCC

Immunotherapy is a novel treatment modality in HNSCC, designed to enhance the immune response to eliminate cancer cells [80]. Nivolumab and pembrolizumab, anti-PD-1 immune checkpoint inhibitors, were approved in 2016 for the treatment of patients with recurrent HNSCC refractory to platinum-based regimens. Most HNSCC patients, however, will experience disease progression when on these agents. Predictive biomarkers are therefore urgently needed to determine which patients will benefit from immunotherapy. Circulating miRNAs and EVs containing miRNAs were already shown to have potential as predictive biomarkers for anti PD-1/PD-L1 treatment response in NSCLC [81] and in HPV-related cervical cancer [82].

Two ongoing clinical trials, currently in the recruitment phase, aim to assess the potential of extracellular miRNAs as predictive biomarkers for immunotherapy in HNSCC. Towards this goal, researchers from VU University Medical Center investigate plasma vesicle miRNAs in patients with locally advanced oral cancer treated with neoadjuvant nivolumab at baseline and after surgery [83]. The study is estimated to complete in April 2022. In the FRAIL-IMMUNE study, estimated to complete in March 2022, researchers will perform plasma miRNA profiling at inclusion and 1, 2, 3, 4, 5, and 12 months after the treatment initiation [84]. The next years will thus uncover the predictive potential of miRNA-based biomarkers for this new and highly promising form of treatment in HNSCC.

## 6. Potential, Challenges and Future Directions

Numerous research studies have shown that miRNA-based biomarkers, of which expression is measured in body fluids such as a serum, plasma, and saliva, have the potential to fill an empty niche in HNSCC diagnostics. They could serve as accurate and minimally invasive tools for patient stratification and disease status monitoring. A growing body of evidence points to the involvement of miRNAs in processes relevant for tumorigenesis, metastasis, and resistance to treatment, establishing their suitability as markers of these phenomena. Unfortunately, efforts of many research groups have thus far not resulted in the translation of any miRNA-based biomarkers to the clinic.

The little overlap between the results of seemingly similar studies and the widespread lack of reproducibility [85] has been explained by methodological challenges and the lack of standardized procedures [86,87], which, together with poor reporting, undermines the validation of miRNAs as biomarkers and their clinical utility. Dharmawardana et al. reviewed methods of circulating miRNAs profiled in head and neck cancer and identified large discrepancies in sample collection, storage, pre-processing, RNA isolation, quantification, and data analysis, including normalization [88]. They recommend that the HNC research community adapts a common protocol for miRNA biomarker discovery or, if reaching a consensus is impossible, that methodology is reported in sufficient detail to allow for reproducibility and meta-analyses of results from multiple studies.

One of the primary reasons for the plethora of miRNAs reported only in one or two studies is selective profiling, where only a few miRNAs, chosen based on literature search or functional analysis of gene targets, are evaluated in patient samples. Comprehensive assessment of whole panels of miRNAs, either based on qPCR or next-generation sequencing, is the first step towards more consistency between research.

Another important issue that should be addressed to establish extracellular miRNAs as the biomarkers of HNSCC metastasis and progression is their source and relation to the cancer tissue. Three main potential routes via which miRNAs may enter the circulation include direct secretion by cancer cells, selective packing into ‘microparticles’, or release in cell-derived exosomes [89]. Extracellular miRNAs for which altered levels are detected in HNSCC patients or for which are predictive of the disease progression may, however, not necessarily originate from the cancer cells. They may be released from the adjacent normal tissue cells, e.g., as a result of tumor-associated inflammation, from the immune and stromal cells of the tumor environment, or even from blood cells [61]. More studies should therefore address the question of the source of miRNAs detected in extracellular fluids. This can be accomplished by comparing the level of miRNAs of interest between cancer and normal tissues or by tracking the origin of EVs that encapsulate miRNAs, e.g., by the proteomic assessment of the tumor tissue and EV protein cargo.

In conclusion, miRNAs in serum, plasma, and saliva remain the particles of high biomarker potential. However, based on the presented inconclusiveness in current research, we must recommend the usage of qPCR-based panels or small RNA sequencing rather than the selection of particular miRNAs for validation in further research. Only this approach can lead to more overlap between the study results and the identification of extracellular miRNA-based biomarkers which could be translated to the clinic to predict disease progression, recurrence, and metastasis in HNSCC.

## Figures and Tables

**Figure 1 ijms-20-04799-f001:**
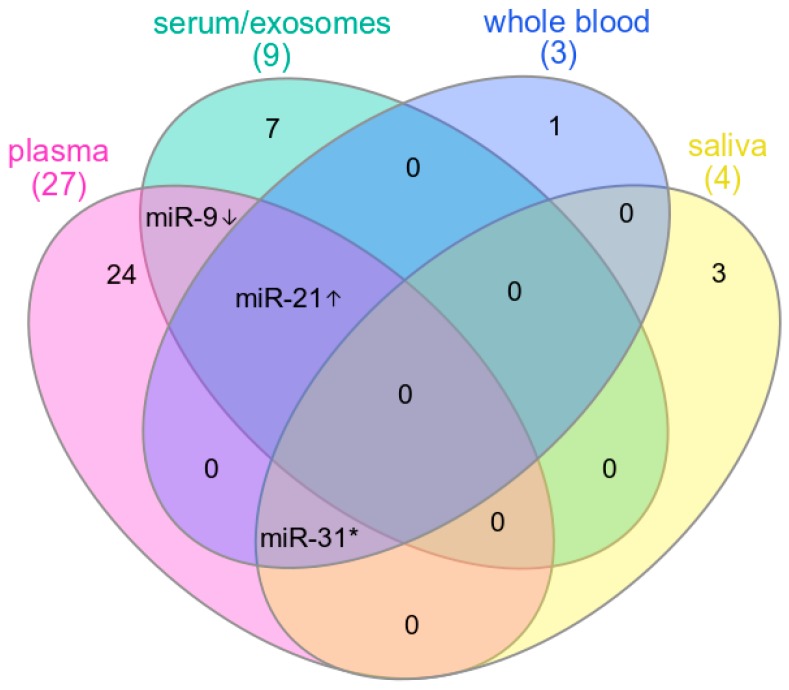
Extracellular miRNAs implied as biomarkers in HNSCC. miRNAs that appeared in studies investigating different biofluids are listed in the figure. **↑**—miRNAs for which higher level was associated with worse prognosis; **↓**—miRNAs for which lower level was associated with worse prognosis; *****—miRNAs with discordant results between studies.

**Table 1 ijms-20-04799-t001:** miRNAs implied as biomarkers in head and neck squamous cell carcinoma (HNSCC).

miRNA	Reference	Cancer Type	Assay Method	Sample Material	Main outcome—Association	Additional Outcomes
miR-9	Lu J et al., 2014 [34]	NPC	qRT-PCR	plasma	lower expression in metastatic patients	lower expressions in patients compared with controls; lower expression associated with advanced stage; higher expression in post-treatment compared to pre-treatment samples
miR-9	Xu X et al., 2018 [35]	NPC	qRT-PCR	plasma	higher expression at posttreatment and lower at recurrence or metastasis	
miR-124	Xu X et al., 2018 [35]	NPC	qRT-PCR	plasma	higher expression at posttreatment and lower at recurrence or metastasis	
miR-892b	Xu X et al., 2018 [35]	NPC	qRT-PCR	plasma	higher expression at posttreatment and lower at recurrence or metastasis	
miR-3676	Xu X et al., 2018 [35]	NPC	qRT-PCR	plasma	higher expression at posttreatment and lower at recurrence or metastasis	
miR-22	Liu N et al., 2014 [36]	NPC	qRT-PCR	serum	higher expression associated with shorter OS	
miR-572	Liu N et al., 2014 [36]	NPC	qRT-PCR	serum	higher expression associated with shorter OS	
miR-638	Liu N et al., 2014 [36]	NPC	qRT-PCR	serum	higher expression associated with shorter OS	
miR-1234	Liu N et al., 2014 [36]	NPC	qRT-PCR	serum	higher expression associated with longer OS	
miR-31	Yi SJ et al., 2019 [37]	NPC	qRT-PCR	whole blood	lower expression associated with local lymph node metastasis, but not distant metastasis	lower expression in patients with NPC than in healthy controls; lower expression associated with worse disease stage and T classification
miR-130b	Severino P et al., 2015 [38]	OSCC	qRT-PCR	plasma	lower expression associated with positive lymph node status	
miR-20b	Severino P et al., 2015 [38]	OSCC	qRT-PCR	plasma	higher expression in patients with lymph node metastases	
miR-106	Severino P et al., 2015 [38]	OSCC	qRT-PCR	plasma	higher expression in patients with lymph node metastases	
miR-296	Severino P et al., 2015 [38]	OSCC	qRT-PCR	plasma	higher expression in patients with lymph node metastases	
miR-301a	Severino P et al., 2015 [38]	OSCC	qRT-PCR	plasma	lower expression in patients with lymph node metastases	
miR-222	Chang YA et al., 2018 [39]	OSCC	qRT-PCR	plasma	lower expression associated with lymph node metastases and higher disease stage; expression decreasing with tumor progression	higher expression in patients with OSCC than in patients with oral leukoplakia
miR-423	Chang YA et al., 2018 [39]	OSCC	qRT-PCR	plasma	lower expression associated with lymph node metastases and higher disease stage; expression decreasing with tumor progression	higher expression in patients with OSCC than in healthy controls and patients with oral leukoplakia
miR-92b	Yan Y et al., 2017 [40]	OSCC	qRT-PCR	plasma	lower expression associated with recurrence	downregulated in in OSCC samples compared with healthy controls
miR-196a	Liu CJ et al., 2012 [41]	OSCC	qRT-PCR	plasma	higher expression in patients with recurrence during follow-up	higher expression in patients with OSCC than in controls
miR-375	Yan Y et al., 2017 [40]	OSCC	qRT-PCR	plasma	lower expression associated with recurrence	downregulated in in OSCC samples compared with healthy controls
miR-486	Yan Y et al., 2017 [40]	OSCC	qRT-PCR	plasma	lower expression associated with recurrence	downregulated in in OSCC samples compared with healthy controls
miR-31	Liu CJ et al., 2010 [42]	OSCC	qRT-PCR	saliva, plasma	lower expression after tumor excision	higher expression in patients with OSCC than in healthy controls; higher expression in saliva than in plasma
miR-483	Xu H et al., 2016 [43]	OSCC	microarray, qRT-PCR	serum	higher expression associated with shorter OS, higher disease stage and lymph node metastases	higher expression in patients with OSCC than in healthy controls
miR-9	Sun L et al., 2016 [44]	OSCC	qRT-PCR	serum	higher expression associated with longer OS and DFS	lower expression in patients with OSCC than in healthy controls; expression associated with stage and lymph node metastasis
miR-626	Shi J et al., 2019 [45]	OSCC	qRT-PCR	serum	higher expression associated with shorter OS	
miR-5100	Shi J et al., 2019 [45]	OSCC	qRT-PCR	serum	higher expression associated with shorter OS	
miR-21	Ren W et al., 2014 [46]	OSCC	qRT-PCR	whole blood	higher expression associated with positive lymph node status	higher expression in patients with OSCC than in healthy controls; lower expression associated with well-differentiated tumor histology
miR-3651	Ries J et al., 2014 [47]	OSCC	qRT-PCR	whole blood	higher expression associated with lymph node metastases, higher tumor grade and stage	higher expression in patients with OSCC than in healthy controls
miR-184	Wong TS et al., 2008 [48]	TSCC	qRT-PCR	plasma	lower expression after the surgical removal of primary tumor	higher expression in patients with tongue SCC than in healthy controls
miR-221	Yilmaz SS et al., 2015 [49]	larynx cancer	qRT-PCR	plasma	expression higher in LC patients than in healthy controls and turning back to normal after tumor excision	
miR-21	Wang J et al., 2014 [50]	LSCC	qRT-PCR	serum-derived exosomes	higher expression associated with higher stage and positive lymph node status	higher expression in patients with LSCC than in patients with vocal cord polyps
miR-139	Duz MB et al., 2016 [51]	TSCC	qRT-PCR	saliva	expression lower in TSCC patients and turning back to normal levels after tumor excision	
miR-21	Hsu CM et al., 2012 [52]	HNSCC	qRT-PCR	plasma	expression turning back to normal after surgery in patients with good prognosis; not changing in patients with poor prognosis	higher expression in patients with HNSCC than in healthy controls
miR-26b	Hsu CM et al., 2012 [52]	HNSCC	qRT-PCR	plasma	expression turning back to normal after surgery in patients with good prognosis; not changing in patients with poor prognosis	
miR-21	Hou B et al., 2015 [53]	HNSCC	qRT-PCR	plasma	expression higher in HNSCC tissue and decreasing in plasma after tumor excision	
miR-99a	Hou B et al., 2015 [53]	HNSCC	qRT-PCR	plasma	expression lower in HNSCC tissue and increasing in plasma after tumor excision	
miR-223	Hou B et al., 2015 [53]	HNSCC	qRT-PCR	plasma	expression higher in HNSCC tissue and decreasing in plasma after tumor excision	expression increased in a patient who relapsed 6 months after operation
miR-142	Summerer I et al., 2015 [54]	HNSCC	qRT-PCR	plasma	higher expression associated with worse locoregional control	higher expression associated with worse PFS
miR-186	Summerer I et al., 2015 [54]	HNSCC	qRT-PCR	plasma	higher expression associated with worse locoregional control	
miR-195	Summerer I et al., 2015 [54]	HNSCC	qRT-PCR	plasma	higher expression associated with worse locoregional control	
miR-374b	Summerer I et al., 2015 [54]	HNSCC	qRT-PCR	plasma	higher expression associated with worse locoregional control	
miR-574	Summerer I et al., 2015 [54]	HNSCC	qRT-PCR	plasma	higher expression associated with worse locoregional control	
miR-93	Greither T et al., 2017 [55]	HNSCC	qRT-PCR	saliva	higher expression 12 months post-radiotherapy than at baseline	expression inversely correlated with salivary gland function
miR-200a	Greither T et al., 2017 [55]	HNSCC	qRT-PCR	saliva	higher expression 12 months post-radiotherapy than at baseline	

NPC—nasopharyngeal carcinoma; OSCC—oropharyngeal squamous cell carcinoma TSCC—squamous cell carcinoma of the tongue; qRT-PCR—quantitative real-time PCR; OS—overall survival; DFS—disease-free survival; PFS – progression-free survival; LSCC—laryngeal squamous cell carcinoma.

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
