# Peer review of "Extracellular miRNAs as Biomarkers of Head and Neck Cancer Progression and Metastasis"

_ijms, 2019, doi:10.3390/ijms20194799_

Round 1

Reviewer 1 Report

The present review article by Nowica et al. is a very well written article, the review is comprehensive with current information. My only suggestion is authors could include some information about the status of clinical trials associated with extracellular miRNA and HNSCCs. There are some minor typographic errors authors should go through the article once again. I recommend its publication

Author Response

We thank both Reviewers for the careful reading of our manuscript and for their helpful comments. We agree with all their suggestions and we made changes in the manuscript accordingly. A point-by-point response to the Reviewer’s comments is provided (Reviewer’s comments in italics):

The present review article by Nowicka et al. is a very well written article, the review is comprehensive with current information.

We thank the Reviewer for the positive feedback.

My only suggestion is authors could include some information about the status of clinical trials associated with extracellular miRNA and HNSCCs.

We included the information about clinical trials associated with extracellular miRNAs and HNSCCs. As the ongoing clinical trials planning miRNA assessment are primarily focused on extracellular miRNAs as predictive biomarkers of the response to immunotherapy with anti PD-1/PD-L1 agents, we decided to incorporate these information in a new section, “miRNAs as predictive biomarkers for immunotherapy in HNSCC” (Section 5: lines 276-293). We believe that this amendment significantly improved the manuscript and we thank the Reviewer for this suggestion.

There are some minor typographic errors authors should go through the article once again. I recommend its publication

We revised the manuscript and fixed any typographic errors that we could find.

Reviewer 2 Report

This is a nicely written review regarding (extracellular) microRNAs and head and neck cancer (progression and metastasis).

I have several suggestions/observations which (I believe) would make this work stand out among the other reviews on this subject:

The critical analysis of the concordance of microRNAs changes in tissues vs whole blood/serum/plasma and saliva is worth expanding, emphasizing which of the "concordant" microRNAs have been shown to have potential clinical value. I find the first two figures somehow redundant, since the information they provide is already present in table1; I think a Venn diagram highlighting the microRNAs common for tissues/blood/plasma/serum/saliva and their association with prognosis would be more helpful. There is ample literature regarding the possible sources of the circulating microRNAs;  this is a perspective worth exploring in relation to HNSCC. Since you have discussed the microRNAs associated with HPV, it would be worth analyzing the influence of other risk factors (tobacco, alcohol). The microRNA target chapter is unexpectedly sketchy and I'm not quite sure I understand the methodology used and why it has been focused only on HPV associated HNSCC. I would suggest either expanding it to a proper, more comprehensive gene network analysis of microRNA targets, either removing it altogether.

Author Response

We thank both Reviewers for the careful reading of our manuscript and for their helpful comments. We agree with all their suggestions and we made changes in the manuscript accordingly. A point-by-point response to the Reviewer’s comments is provided (Reviewer’s comments in italics):

This is a nicely written review regarding (extracellular) microRNAs and head and neck cancer (progression and metastasis).
I have several suggestions/observations which (I believe) would make this work stand out among the other reviews on this subject:

We thank the Reviewer for the positive feedback.

The critical analysis of the concordance of microRNAs changes in tissues vs whole blood/serum/plasma and saliva is worth expanding, emphasizing which of the "concordant" microRNAs have been shown to have potential clinical value.

We agree with the Reviewer that adding more information about miRNA differences concordant between tissues and extracellular fluids would be beneficial for the interpretability of our review. We therefore added such information for miR-21 (lines 123-125), miR-31 (lines 148-150) and the information about lack of significant correlation between plasma and tissue levels of several therapy-responsive miRNAs found by Summerer et al (lines 265-267). However, given the wide variety of study designs, methods and materials used, there is little consistency between findings of different study groups. It is therefore, in our opinion, the main purpose of this review to highlight these discrepancies to urge scientists working in this field to identify gaps in knowledge and strive towards standardization of the methods and reporting styles.

I find the first two figures somehow redundant, since the information they provide is already present in table1; I think a Venn diagram highlighting the microRNAs common for tissues/blood/plasma/serum/saliva and their association with prognosis would be more helpful.

We replaced those figures with a Venn diagram representing miRNAs common for blood/plasma/serum/saliva as suggested.

There is ample literature regarding the possible sources of the circulating microRNAs;  this is a perspective worth exploring in relation to HNSCC.

A paragraph regarding possible sources of the circulating miRNAs was added (lines 345-352).

Since you have discussed the microRNAs associated with HPV, it would be worth analyzing the influence of other risk factors (tobacco, alcohol).

We agree with the Reviewer that the association of extracellular miRNA levels with established risk factors of HNSCC, such as smoking and alcohol consumption, is worth exploring. However, we did not find any studies that would identify significant relationship between those factors and extracellular miRNA levels in HNSCC. We added this information to our manuscript (lines 231-235).

The microRNA target chapter is unexpectedly sketchy and I'm not quite sure I understand the methodology used and why it has been focused only on HPV associated HNSCC. I would suggest either expanding it to a proper, more comprehensive gene network analysis of microRNA targets, either removing it altogether.

In order to comprehensively address the issue of miRNA-mRNA interactions we would have to expand the manuscript substantially to encompass issues such as tissue types, treatment interactions and different bioinformatics predictive tools estimating the strength of miRNA binding. Therefore, given the size limitations of the manuscript and its intent to be a review rather than a bioinformatics paper, we decided to remove this section as per Reviewer’s suggestion alongside appendix A.

Round 2

Reviewer 2 Report

Overall, nice work: congratulations!